# A circadian clock drives behavioral activity in Antarctic krill (*Euphausia superba*) and provides a potential mechanism for seasonal timing

Lukas Hüppe[1,2]*, Dominik Bahlburg[2], Ryan Driscoll[3], Charlotte Helfrich-Förster[1], Bettina Meyer[2,4,5]*

[1]Neurobiology and Genetics, University of Würzburg, Biocenter, Theodor-Boveri-Institute, Würzburg, Germany; [2]Section Polar Biological Oceanography, Alfred Wegener Institute Helmholtz Centre for Polar and Marine Research, Bremerhaven, Germany; [3]National Oceanography Centre, European Way, Southampton, United Kingdom; [4]Institute for Chemistry and Biology of the Marine Environment, University of Oldenburg, Oldenburg, Germany; [5]Helmholtz Institute for Functional Marine Biodiversity at the University of Oldenburg (HIFMB), Oldenburg, Germany

*For correspondence:
lukas.hueppe@uni-wuerzburg.
de (LH);
bettina.meyer@awi.de (BM)

Competing interest: The authors declare that no competing interests exist.

## eLife Assessment

This **important** study substantially advances our understanding of the circadian clock in Antarctic krill, a key species in the Southern Ocean ecosystem. Through logistically challenging shipboard experiments conducted across seasons, the authors provide **compelling** evidence for their conclusions. The study will be of broad interest to marine biologists and ecologists.

**Abstract** Antarctic krill is a species with fundamental importance for the Southern Ocean ecosystem. Their large biomass and synchronized movements, like diel vertical migration (DVM), significantly impact ecosystem structure and the biological carbon pump. Despite decades of research, the mechanistic basis of DVM remains unclear. Circadian clocks help organisms anticipate daily environmental changes, optimizing adaptation. In this study, we used a recently developed activity monitor to record swimming activity of individual, wild-caught krill under various light conditions and across different seasons. Our data demonstrate how the krill circadian clock, in combination with light, drives a distinct bimodal pattern of swimming activity, which could facilitate ecologically important behavioral patterns, such as DVM. Rapid damping and flexible synchronization of krill activity indicate that the krill clock is adapted to a life at high latitudes and seasonal activity recordings suggest a clock-based mechanism for the timing of seasonal processes. Our findings advance our understanding of biological timing and high-latitude adaptation in this key species.

## Introduction

Antarctic krill (*Euphausia superba,* hereafter referred to as krill) is a pelagic, up to 6 cm long crustacean, which is endemic to the Southern Ocean. With an estimated biomass of 300–500 million tons (*Atkinson et al., 2009*), krill is among the most abundant wild species worldwide (*Bar On et al., 2018*) and an important prey for predators such as whales, seals, and penguins (*Trathan et al., 2016*). Its huge abundance and central position in the food web makes krill a key species of fundamental

**eLife digest** The Southern Ocean is home to whales, seals, seabirds and other iconic wildlife. All of these animals depend either directly, or indirectly, on a small marine prey species known as Antarctic krill, which thrives in the harsh conditions of the Southern Ocean. At night, large swarms of krill move towards the water surface to feed on plankton before returning to the depths during the day to avoid whales, fish and other predators. This synchronized movement influences the structure of the ecosystem in a number of ways by transporting carbon and influencing predator-prey interactions.

Researchers have observed the movements of krill swarms for many decades, but the processes controlling this swimming behavior remained unknown, in part, due to a lack of tools that can track the movements of individual krill. Do the krill simply respond to light and other external cues, or do they also have internal biological clocks that can maintain the observed rhythms even without such cues?

In 2024, researchers developed a new monitor known as AMAZE, which can record the swimming activity of individual krill in tanks of seawater. Hüppe et al. – including many of the researchers involved in the 2024 work – have now used this technique to trace the movement of individual wild-caught krill under different light conditions and seasons.

Hüppe et al. captured krill from the Southern Ocean on a commercial fishing boat and transferred them into a tank on the vessel for experiments. Observations revealed that the krill were most active at night, matching their natural patterns of migration in the wild. These patterns of nighttime activity adjusted to the changing length of the night over the seasons. Furthermore, the krill maintained a daily rhythm of activity even when they were kept in constant darkness for several days.

These findings suggest that an internal biological clock, in combination with light cues, regulates the swimming patterns of krill and helps them adapt to daily and seasonal changes in their environment. Understanding these internal rhythms will be key to assessing how well krill may cope with rapid changes in their environment due to climate change, which are particularly pronounced in polar regions.

importance for the functioning of the Southern Ocean ecosystem (*Cavan et al., 2019*; *Trinh et al., 2023*).

Krill is mostly found in swarms, the largest of which can be hundreds of meters long and tens of meters in height, likely reflecting a behavioral adaptation to minimize the risk of predation (*Tarling et al., 2009*; *Hamner and Hamner, 2000*). Along with their pronounced swarming behavior, krill display diel vertical migration (DVM), a common behavior among many pelagic zooplankton organisms across different taxa found in all oceans (*Bandara et al., 2021*). DVM is commonly characterized by an ascent to the surface layers in the dark of the night to feed and a descent to darker parts of the water column during the day to avoid visually hunting predators (*Bandara et al., 2021*; *Hays, 2003*). The regular, synchronized movement of this large biomass significantly shapes the structure of ecosystems (*Nichols et al., 2022*) and has large impacts on biogeochemical cycles (*Aumont et al., 2018*; *Archibald et al., 2019*). This includes the recycling of nutrients, stimulation of primary production, as well as transport and storage of carbon in the deep ocean (*Cavan et al., 2019*). Although such a 'regular' DVM pattern has been commonly observed for krill swarms (*Godlewska, 1996*; *Everson, 1983*), there is considerable behavioral variation, including reverse DVM, higher frequency migrations, or no synchronized migrations at all (*Godlewska, 1996*; *Bahlburg et al., 2023b*). While the adaptive advantage of balancing increased predation pressure during the day with the need to feed seems obvious, the underlying mechanisms of DVM and its modulations are not understood. Light is thought to be the most reliable and thus important environmental cue for DVM (*Ringelberg and Van Gool, 2003*). However, a study in the marine copepod *Calanus finmarchicus* indicates that DVM behavior is underpinned by a circadian clock, which helps the animals to adjust behavior, physiology, and gene expression to the day-night cycle (*Häfker et al., 2017*). Similarly, a study on Norwegian krill, *Meganyctiphanes norvegica*, suggested swimming activity in krill is under clock control, with increased activity during the dark phase (*Velsch and Champalbert, 1994*). In Antarctic krill, similar studies have been conducted to test the hypothesis of a clock involvement in DVM behavior (*Gaten et al., 2008*; *Piccolin et al., 2020*). While these studies gave first hints that daily behavioral activity

in krill is partly under endogenous control, a large variability hindered a clear characterization of the circadian behavioral activity of krill.

Previous studies on endogenous rhythms in krill identified the molecular components of a krill circadian clock (*Biscontin et al., 2017*; *Shao et al., 2023*). It has further been shown that the krill circadian clock drives daily rhythms in gene expression, metabolism, and behavior (*Piccolin et al., 2020*; *Teschke et al., 2011*; *Biscontin et al., 2019*). Part of krill's success lies in its strong adaptation to the extreme seasonality of the Southern Ocean environment (*Ducklow et al., 2007*), which is reflected in a fundamental, seasonal regulation in body composition, metabolic activity, feeding, growth (*Meyer et al., 2010*; *Quetin and Ross, 1991*), sexual maturity (*Kawaguchi et al., 2007*; *Siegel, 2012*), and gene expression (*Seear et al., 2012*; *Höring et al., 2021*). Results from laboratory studies showed that photoperiod is a major driver for seasonal ecophysiological changes in krill (*Brown et al., 2011*; *Höring et al., 2018*; *Teschke et al., 2007*), and it has been suggested that an endogenous timekeeping system (i.e. biological clock) is involved, which uses photoperiod as a synchronizing cue (*Höring et al., 2018*; *Piccolin et al., 2018b*).

Circadian clocks are widespread and well-studied endogenous timing mechanisms which allow an organism to fine-tune biological functions to specific times of the day-night cycle. The endogenous oscillation is produced by transcriptional-translational feedback loops of a set of clock genes and their translation products (*Tomioka and Matsumoto, 2010*), which, under the absence of external cues, produce a temporal signal with a period ($\tau$) of about 24 hr. Under natural conditions, the rhythm of the central oscillator is synchronized to exactly 24 hr by predictable environmental cycles, called *zeitgeber*, of which light is usually the most important (*Roenneberg et al., 2003*). The rhythmic signal of the central oscillator is superimposed on downstream processes, which results in the temporal regulation of several biological functions, such as gene expression, physiology, and behavior. Thus, the circadian clock allows organisms to anticipate daily environmental cycles and adapt their biology, ultimately increasing survival and fitness (*Krittika and Yadav, 2020*). Since the circadian clock implicitly measures day length, it is also capable of providing seasonal information. Consequently, the circadian clock has also been shown to be involved in the timing of seasonal functions, such as the induction of reproduction or diapause (*Saunders, 2020*; *Helfrich-Förster, 2024*).

While most knowledge about circadian clocks and their involvement in organism's daily and seasonal processes stems from terrestrial model organisms, little is known about clocks in the marine environment (*Tessmar-Raible et al., 2011*). This is especially true for marine habitats in polar regions, characterized by extreme photic conditions, including polar night and midnight sun phases. An understanding of the mechanistic underpinnings of biological timing is essential to comprehend how organisms adapt to their specific environment. This is particularly important in times of rapid environmental change, which can alter the temporal synchronization of trophic interactions, including DVM, feeding, and swimming behavior, and, thus, the functioning of whole ecosystems (*Helm et al., 2013*).

In this study, we use the novel Activity Monitor for Aquatic Zooplankter (AMAZE) (*Hüppe et al., 2024*) to investigate the mechanistic basis of swimming activity of wild-caught Antarctic krill under controlled environmental conditions onboard a commercial krill fishing vessel. We provide novel evidence that the circadian clock underlies ecologically important behaviors, such as DVM, and discuss how the krill circadian clock could control the timing of seasonal life cycle functions in krill.

## Results

Two sets of experiments were conducted for this study (*Figure 1*). In the first set of experiments (*experiment 1*) we adopt a standard chronobiological experiment design, where we monitor krill behavioral activity under simulated short days (*short-day treatment*) or long days (*long-day treatment*), before they are transferred into constant darkness for several days to investigate the influence of a circadian clock on krill behavior. In a second set of experiments (*experiment 2*) krill vertical migration behavior in the field was recorded for several days before sampling and transferring krill into constant darkness. This experiment was conducted in four seasons (*summer treatment*, *late summer treatment*, *autumn treatment*, and *winter treatment*).

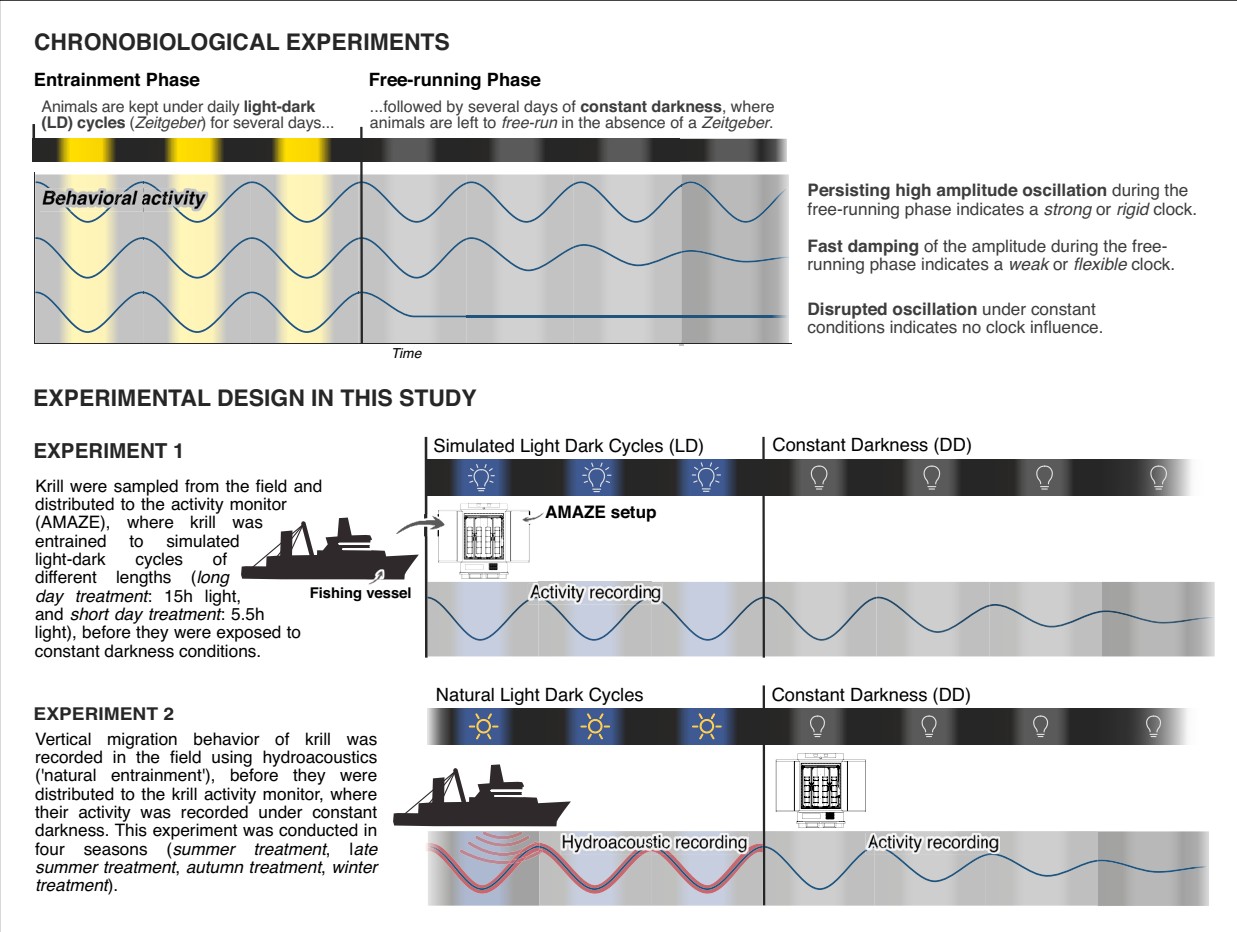

**Figure 1.** Experimental design of chronobiological studies. Overview of the basic principle of chronobiological experiment design and how it is adopted in our study to characterize the influence of the circadian clock on swimming activity of wild-caught Antarctic krill.

The online version of this article includes the following figure supplement(s) for figure 1:

**Figure supplement 1.** Locations of krill sampling and hydroacoustic recordings for behavioral experiments.

## A circadian clock is involved in the daily regulation of swimming activity

To investigate whether a circadian clock is involved in regulating daily rhythms of swimming activity, we sampled krill from the field under short-day conditions in winter (*short-day treatment*, sampling date: May 31, 2021, local photoperiod: 5.5 hr) and under long-day conditions in late summer (*long-day treatment*, sampling date: February 16, 2022, local photoperiod: 15.3 hr; see *Table 1* for details). We exposed 11 and 9 sampled individuals to 3 days of light-dark conditions in the activity monitor simulating short-day conditions and to 5 days of LD simulating long-day conditions, respectively. The photoperiod in the activity monitor approximated the natural photoperiod in the field during sampling. The period of LD was followed by 5 days of constant darkness. The data under LD conditions show that krill swimming activity at both individual and group level increased during the dark phase and showed a strong synchronization with the light-dark cycle provided (*Figure 2—figure supplements 1 and 2*).

As synchronization between individuals and robustness of activity patterns under DD within individuals was strongest in krill from the short-day treatment, we used these data for initial assessments of circadian activity rhythms. Under constant darkness conditions in the short-day treatment, krill swimming activity continued in several individuals (n=6; 54%) with significant circadian rhythmicity over the 5 days of DD conditions (*Figure 2a and b*, *Figure 2—figure supplement 3*), indicating the involvement of a circadian clock in the daily regulation of swimming activity. Nevertheless, in most

**Table 1.** Metadata for behavioral experiments.

For each experiment (Experiment ID), the metadata show the light regime during the experiment in the activity monitor (LD: light-dark cycle, DD: constant darkness), the duration of the experiments under the respective light condition, the local date and time, region (BS: Bransfield Strait, SOI: South Orkney Islands), exact location, and depth for krill sampling, the local natural photoperiod at the time of krill sampling, the number of krill used for each experiment, and their mean length and standard deviation.

| Experiment ID | Experimental light regime | Experiment duration (days) | Sampling date (UTC-2) | Sampling time (UTC-2) | Sampling region | Sampling location (latitude °S/longitude °W) | Sampling depth (m) | Natural photoperiod (hr) | n Krill (total/ male/ female) | Mean ±SD length (mm) |
|---|---|---|---|---|---|---|---|---|---|---|
| Experiment 1 (short-day treatment) | LD-DD | 3 (LD), 5 (DD) | 2021-05-31 | 13:02 | BS | 62.95/57.77 | 170 | 5.5 | 11/5/6 | 45±2.6 |
| Experiment 1 (long-day treatment) | LD-DD | 5 (LD), 5 (DD) | 2022-02-16 | 15:50 | SOI | 60.04/46.45 | 94 | 15.3 | 9/4/5 | 44.4±3.7 |
| Experiment 2 (summer treatment) | DD | 8 | 2022-01-22 | 17:03 | SOI | 60.08/46.46 | 157 | 17.5 | 9/8/1 | 42.4±1.5 |
| Experiment 2 (late summer treatment) | DD | 6 | 2022-03-01 | 11:00 | SOI | 60.0/44.73 | 86.5 | 14.0 | 9/4/5 | 48.9±4.4 |
| Experiment 2 (autumn treatment) | DD | 7 | 2022-03-15 | 08:40 | SOI | 60.28/46.43 | 119.2 | 12.6 | 10*/4/5 | 51.3±1.6* |
| Experiment 2 (winter treatment) | DD | 4 | 2021-06-10 | 12:56 | SOI | 60.5/46.11 | 185 | 5.9 | 9/2/7 | 49.2±3.9 |

*Length and sex of one individual undetermined.

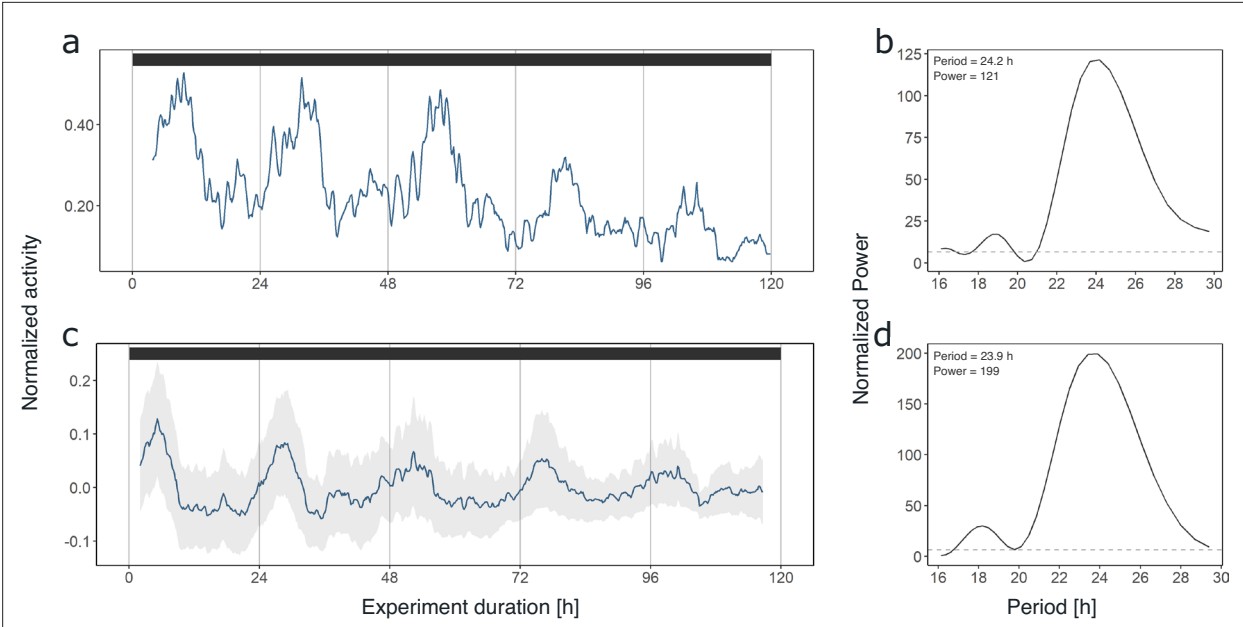

**Figure 2.** Krill swimming activity persists under constant conditions. Swimming activity under 5 days of DD conditions in experiment 1 of individual #9 after entrainment to simulated short-day conditions (**a**) and group mean swimming activity of rhythmic individuals under 5 days of DD conditions from the same experiment (n=6, **c**). The result from Lomb-Scargle periodogram (LSP) analysis shows significant circadian rhythmicity of swimming activity of individual #9 (period: 24.2 hr, **b**) as well as for group mean swimming activity (period: 23.9 hr, **d**) during DD conditions of experiment 1 (short-day treatment). Gray shading represents standard error of the mean (s.e.m.). Color bars at the top indicate light conditions (constant darkness).

The online version of this article includes the following figure supplement(s) for figure 2:

**Figure supplement 1.** Krill group activity under LD conditions.

**Figure supplement 2.** Swimming activity of individual krill under LD conditions.

**Figure supplement 3.** Swimming activity of individual krill under DD conditions.

individuals, the amplitude of the rhythm decreased over time (*Figure 2a*, Figure 4a, *Figure 2—figure supplement 3*). The patterns of swimming activity observed at the individual level were reflected at the group level, showing a strong synchronization of increased swimming activity with the dark phase under light-dark conditions (*Figure 3a*, *Figure 2—figure supplement 1*), and persistent significant circadian swimming activity under constant conditions with a continuously decreasing amplitude throughout the experiment (*Figure 2c and d*; *Figure 4a*: paired t-test, p-value=0.005, *Figure 4c*: paired t-test, p-value=0.012).

## Light affects the amplitude and phase of swimming activity

Comparing the mean swimming activity of rhythmic individuals over an average day from the short-day and long-day treatments under LD conditions reveals a clear synchronization of swimming activity with the dark phase (*Figure 3*), visible in the sharp increase in activity around the times of lights-off and decrease at lights-on. Furthermore, the time of increased activity during the dark phase exhibits a discernible pattern comprising three distinct activity bouts (*Figure 3*). First, an increase in activity is observable in the evening (i.e. end of illumination), followed by a decrease in activity toward midnight. A second increase in swimming activity is observable in the latter half of the dark phase, followed by a decline in activity just before morning. A third short activity bout is observed in the morning (i.e. onset of illumination). This distinctive activity pattern during the dark phase is observed regardless of the photoperiod, resulting in a compression of the three activity bouts in short nights, while they spread out in long nights (*Figure 3*), demonstrating that krill can entrain to the long and short days provided. Furthermore, comparing the activity pattern during short-day LD conditions with the respective activity pattern under DD conditions reveals the persistence of evening and late-night activity increases under constant darkness conditions. In contrast, the morning activity bout is not visible (*Figure 3—figure supplement 1*). This suggests that the circadian clock drives a distinct bimodal

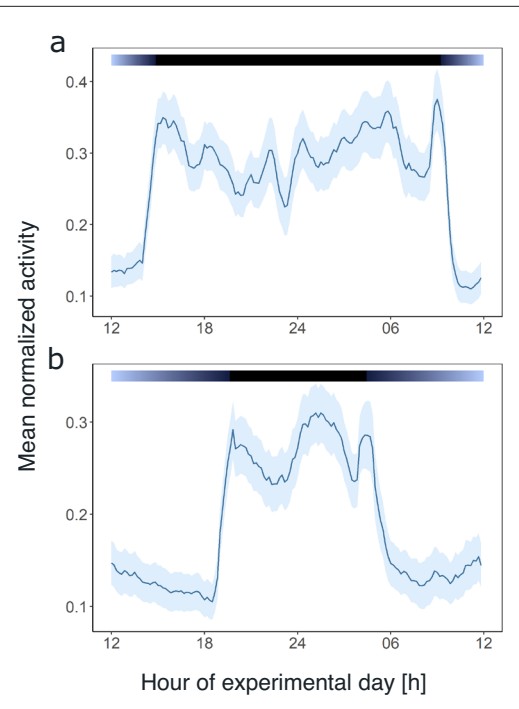

**Figure 3.** Light and the circadian clock shape the daily activity profile of krill. Average day analysis of the group mean swimming activity during experiment 1 over 3 days of short-day simulations (short-day treatment, n=11, **a**) and 5 days of long-day simulations (long-day treatment, n=9, **b**). Color bars at the top indicate light regime under day-night simulations. Shading around the line represents the s.e.m. Data in (**b**) from **Hüppe et al., 2024**.

The online version of this article includes the following figure supplement(s) for figure 3:

**Figure supplement 1.** Comparison of daily swimming activity between LD and DD conditions.

activity pattern with two activity peaks in one day, i.e., the evening and late-night activity bouts. In contrast, the morning activity bout is triggered by the onset of illumination in the experimental setup. Furthermore, the amplitude of swimming activity is higher under LD than DD conditions (**Figure 4B**, Mann-Whitney U-test: p-value<0.001; **Figure 4D**, p-value<0.001).

## Circadian swimming activity persists with a stable phase after entrainment to a wide range of natural photoperiods

To gain insights into potential seasonal differences in the regulation of krill swimming activity, we sampled krill from the field in summer (summer treatment), late summer (late summer treatment), autumn (autumn treatment), and winter (winter treatment) and recorded their swimming activity for 4–8 days under constant darkness conditions, without previous entrainment in the activity monitor (see **Table 1** for details).

Similar to our previous analysis of individual krill swimming activity under constant darkness conditions, a large proportion of the recorded individuals in each experiment (44–100%) exhibited significant rhythmicity in the circadian range (**Figure 5—figure supplements 1–4**), which is also reflected in rhythmic swimming activity at the group level (**Figure 5a–d**, **Figure 5—figure supplement 5**). Interestingly, the pattern of circadian swimming activity, with two peaks during the dark phase, appears to become clearer and higher in amplitude with shortening photoperiods toward winter (**Figure 5a–d**). This is supported by rhythm analysis of the group behavior, which revealed a higher power and a period estimation closer to 24 hr with shortening photoperiod (**Figure 5—figure supplement 6**). While the morning peak is mainly visible in autumn and winter, the late-night activity peak is visible throughout the year, at the same time of the subjective 24 hr-day (**Figure 5e–h**). This pattern appears irrespective of the natural photoperiod the animals were entrained to in their natural environment before they were sampled for the respective experiment.

## Synchronized DVM in the field is present across a wide range of natural photoperiods

To investigate whether krill swarms exhibited daily behavioral patterns in swimming behavior in the field before they were sampled for seasonal experiments, hydroacoustic data were recorded from the fishing vessel, continuously over a 3-day period prior to sampling for the seasonal experiments described above (for vessel positions during hydroacoustic recordings, see **Figure 1—figure supplement 1**). The data provide information about the vertical distribution of krill swarms in the upper water column (<220 m) below the vessel over time (**Figure 6a–d**). The hydroacoustic data in summer, late summer, and autumn show that krill swarms performed DVM, highly synchronized with the local light regime (**Figure 6a–c**). The swarms regularly ascended to the surface layers (<50 m) around sunset and returned to deeper layers (>100 m) around sunrise. In winter, little signal is visible in the upper water layer, indicating very low density or the absence of krill swarms (**Figure 6d**). Consequently, krill for the

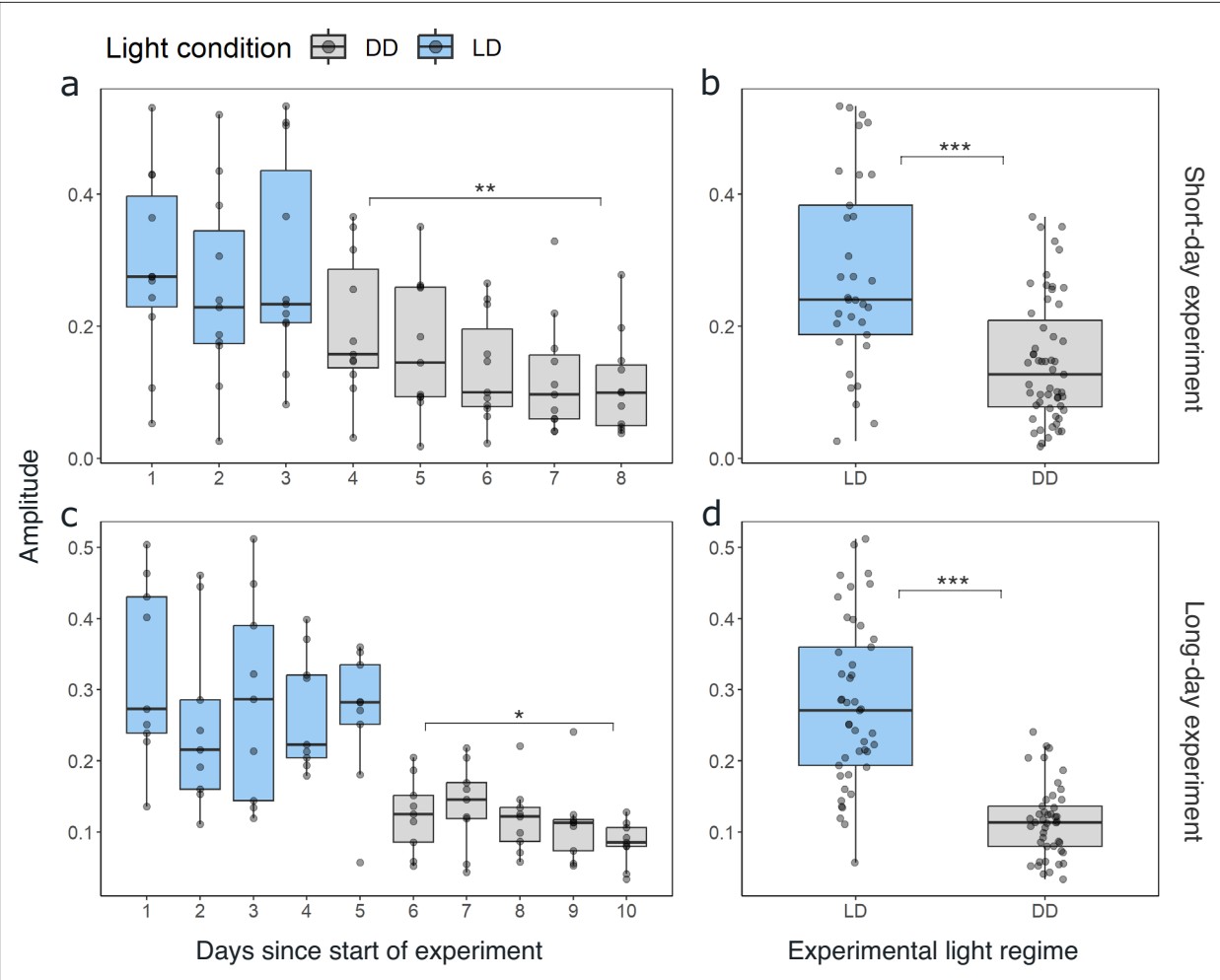

**Figure 4.** The amplitude of swimming activity decreases in DD conditions. Distribution of the daily amplitudes of swimming activity visualized with boxplots for every day of experiment 1 for short-day (**a**) and long-day treatments (**b**). Differences in the daily amplitude of swimming activity between LD and DD conditions for short-day (**b**) and long-day (**d**) treatments. Lower and upper hinges of the boxes correspond to the 25th and 75th percentiles, the upper and lower whiskers extend to the largest and smallest value no further than 1.5 of the interquartile range, respectively, the horizontal line shows the median. Points represent daily amplitudes of individuals for each day and light condition. Differences in amplitude between first and last day under DD tested with paired t-test for short-day (a; n=11, t=3.197, df = 10, p-value=0.005) and long-day treatments (c, n=9, t=2.731, df = 8, p-value=0.012), and between LD and DD conditions with Mann-Whitney U-test for experiment 1 (b; n=88, p-value=<0.001) and experiment 2 (d; n=90, p-value<0.001). Significance levels: p<0.05: *, p<0.01: **, p<0.001: ***.

winter treatment was sampled from deeper waters (185 m). This agrees with observations that krill shift their distribution to deeper waters in winter, where krill have been observed to perform DVM also during this time of year (**Bahlburg et al., 2023b**). The available data only covers the upper 220 m of the water column, meaning that the dynamics of swarms below this depth are missed.

## Discussion

Our study revealed continued rhythmic swimming activity under constant darkness, showing the involvement of a circadian clock in swimming behavior of krill, as has been previously shown for other marine pelagic crustaceans (**Häfker et al., 2017**; **Velsch and Champalbert, 1994**; **Cohen and Forward, 2005**). To date, only three studies have investigated the involvement of a circadian clock in the regulation of swimming behavior in krill species, including *E. superba* (**Gaten et al., 2008**; **Piccolin et al., 2020**) and *M. norvegica* (**Velsch and Champalbert, 1994**). Data from **Piccolin et al., 2020**, showed a strong damping of the amplitude and indication of a remarkably short (~12 hr) free running period (FRP) of vertical swimming behavior of a group of krill under constant darkness (**Piccolin et al.,**

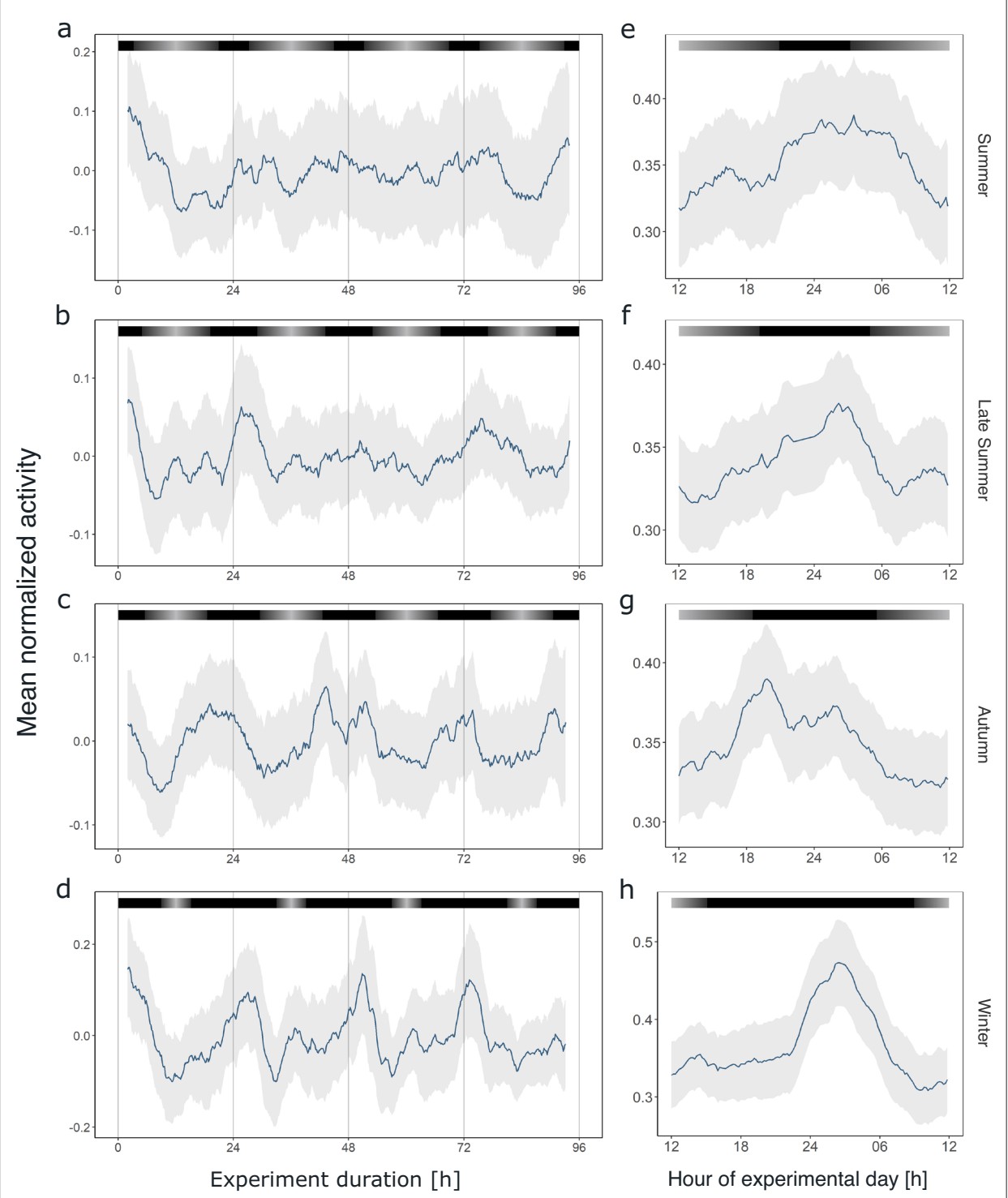

**Figure 5.** Circadian swimming activity persists across seasons. Group mean swimming activity of rhythmic individuals during the first 4 days under DD conditions of experiment 2 for krill sampled for the summer (n = 5, **a**), late summer (n = 8, **b**), autumn (n = 10, **c**), and winter treatments (n = 4, **d**), as well as corresponding average day analysis of the first 4 days of experiment 2 in summer (**e**), late summer (**f**), autumn (**g**), and winter (**h**). Gray shading represents the s.e.m. Color bars at the top indicate the natural photoperiod at the day of sampling.

The online version of this article includes the following figure supplement(s) for figure 5:

**Figure supplement 1.** Individual swimming activity under DD conditions of krill sampled in summer.

**Figure supplement 2.** Individual swimming activity under DD conditions of krill sampled in late summer.

*Figure 5 continued on next page*

*Figure 5 continued*

**Figure supplement 3.** Individual swimming activity under DD conditions of krill sampled in autumn.

**Figure supplement 4.** Individual swimming activity under DD conditions of krill sampled in winter.

**Figure supplement 5.** Group swimming activity under DD conditions across seasons.

**Figure supplement 6.** Rhythm analysis of group swimming activity under DD conditions across seasons.

*2020*). The short period found in *Piccolin et al., 2020*, is in line with our findings of a bimodal activity pattern under DD conditions on the individual level, suggesting that the ~12 hr rhythm in group swimming behavior in *Piccolin et al., 2020*, could have resulted from a bimodal activity pattern at the individual level, as found in our study. Bimodal locomotor activity rhythms are well known across different species, possibly most prominently in the fruit fly (*Drosophila melanogaster*), where two coupled endogenous oscillators control morning and evening activity bouts (*Helfrich-Förster, 2001*). In our experiments, we observed a strong weakening of the evening activity under constant conditions,

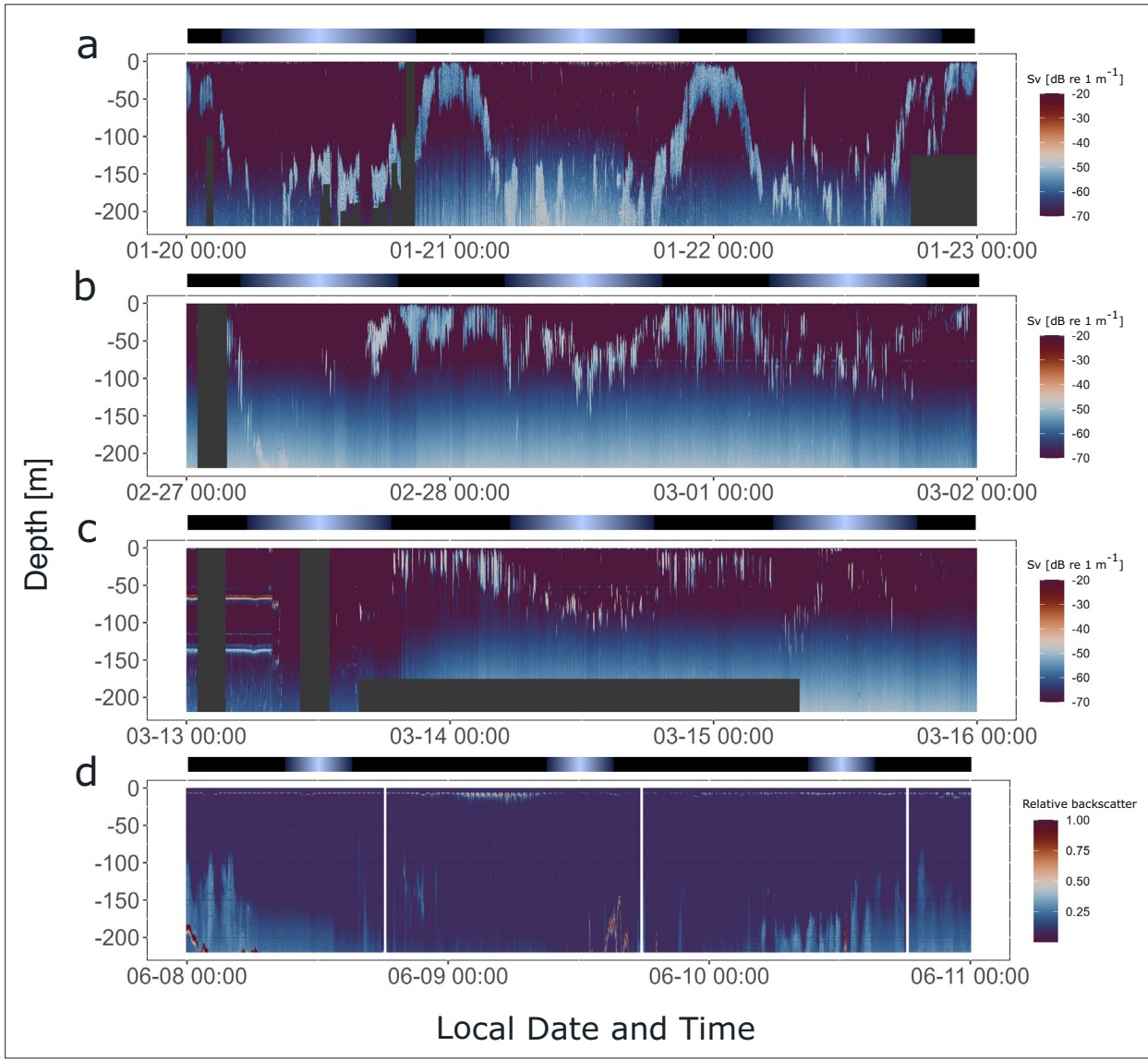

**Figure 6.** Krill display synchronized diel vertical migration (DVM) in the field across a wide range of photoperiods. Hydroacoustic recordings showing the vertical distribution of krill swarms in the upper water column (<220 m) below the vessel, visualized by the mean volume backscattering signal (200 kHz), on the 3 days prior to krill sampling for experiment 2, in summer (**a**), late summer (**b**), autumn (**c**), and winter (**d**). Diffuse blue shading between ~100 and 220 m represents instrument noise. Gray areas depict missing data. Color bars at the top indicate the natural light regime.

especially after entrainment to long days, while the late-night activity peak appeared more robust. The differential variation in the morning and late-night activity suggests that the circadian activity bouts could also be controlled separately at the neuronal level in krill.

Comparing the daily pattern of swimming activity under LD conditions with those under DD, it is clear that the amplitude of the rhythm was significantly higher under LD than under DD. In addition, the activity under LD was organized into three activity phases, one in the evening, one late at night, and a smaller one in the morning. In contrast, under DD there were a maximum of two activity phases during the subjective night. Light is known to play a dual role in regulating daily behavioral activity. On the one hand, light functions as a *zeitgeber*, which synchronizes the circadian clock with the day-night cycle, a process called *entrainment* (*Aschoff, 1960*).

On the other hand, it can directly affect clock output functions such as physiology and behavior, known as *masking* (*Aschoff, 1960*; *Mrosovsky, 1999*). The masking response depends on the activity type of the organism, and light was shown to promote activity in diurnal animals while inhibiting activity in nocturnal ones (*Aschoff and von Goetz, 1989*; *Aschoff and von Goetz, 1988*). Krill are regularly observed performing nocturnal DVM (*Godlewska, 1996*; *Everson, 1983*), including in the present study, with higher activity during the night, including the times of twilight, related to foraging and feeding activity and vertical ascend and descend movements (*Zhou and Dorland, 2004*; *Klevjer and Kaartvedt, 2011*). The observed effect of light on reducing krill swimming activity under LD conditions in our study is thus in line with previous concepts of the exogenous effects of light on nocturnal organisms (*Aschoff, 1960*; *Mrosovsky, 1999*). However, under LD conditions, the observed swimming activity pattern in krill is not solely driven by light. The increase in activity in the evening and the second half of the dark phase under LD conditions flexibly adjusts to the length of the night and seems to persist under constant darkness, suggesting these bouts to be under clock control, as discussed above.

The reduced amplitude in swimming activity after the switch to DD conditions and further damping of amplitude over time under constant darkness has been found in other studies in krill (*Piccolin et al., 2020*) and other marine zooplankton organisms (*Häfker et al., 2017*; *Cohen and Forward, 2005*). A rapidly damping output signal under constant conditions could indicate a weak, low amplitude circadian oscillator, where the damping amplitude of the clock output signal is caused by a rapid desynchronization of clock neurons, when zeitgeber signals are absent (*Webb et al., 2012*). Interestingly, weak oscillators have been found in different fruit fly species (*Bertolini, 2019*; *Beauchamp et al., 2018*), the Svalbard reindeer (*Arnold et al., 2018*), and Svalbard ptarmigan (*Appenroth et al., 2021*), native to high latitudes and species that exhibit pronounced seasonal rhythmicity, such as aphids (*Beer et al., 2017*; *Barberà et al., 2017*). It is known that weaker clocks are more flexible and can be entrained more efficiently than those that are more rigid and self-sustained (*Webb et al., 2012*; *Abraham et al., 2010*). This could confer a significant adaptive advantage to species inhabiting environments characterized by extreme photic conditions (*Bertolini, 2019*; *Beauchamp et al., 2018*; *Bloch et al., 2013*), such as phases of polar night or midnight sun as well as rapid changes in day length, or species that rely on precise photoperiodic time measurement for accurate seasonal adaptation. In the Southern Ocean, photoperiod becomes extreme around the solstices, and its daily change is rapid around the spring and autumn equinoxes. A weak oscillator could, therefore, ensure that krill remains synchronized with the extreme and rapidly changing photic conditions and, thus, its environment.

Indeed, our data from activity recordings under DD conditions of krill sampled in different seasons indicate persistent circadian swimming activity, irrespective of the season. Our findings suggest that the krill circadian clock remains functional and drives rhythmic behavioral output in natural conditions throughout the year. This finding is complemented by the hydroacoustic recordings in the days preceding krill sampling for activity experiments. In summer, late summer, and autumn, krill swarms exhibited apparent DVM behavior, synchronized to the natural photoperiod. While we did not observe DVM in the upper water column during sampling in winter, acoustic recordings from other studies in winter indicate that krill may perform DVM also in winter (*Cisewski et al., 2010*), potentially in deeper layers.

Our data show that light acts in concert with, and is likely modulated by, the circadian clock to achieve the observed daily regulation in krill swimming activity. Nevertheless, further studies at the behavioral, molecular, and neuronal levels are required to elucidate the endogenous and exogenous

effects of light and the circadian clock on the different behavioral aspects observed and to gain further insights into the characteristics of the molecular mechanism of the krill clock.

In a high-latitude region, such as the Southern Ocean, the timing of crucial life history functions to seasonal fluctuations of light, food availability, and sea ice extent is key to organisms' success. In contrast to early studies, which suggested that only food availability was the main driver of krill's seasonal physiological functions (*Ikeda and Dixon, 1982*), later studies have revealed that photoperiod, likely in concert with circannual timing, drives the seasonal regulation of key life cycle functions (*Meyer et al., 2010*; *Brown et al., 2011*; *Höring et al., 2018*; *Teschke et al., 2007*; *Piccolin et al., 2018b*).

Although the exact molecular mechanism of seasonal timing remains unknown, there is clear evidence that the circadian clock measures day length and provides information about the seasonal progression (*Saunders, 2020*; *Helfrich-Förster, 2024*). Several studies in insects (*Saunders, 2024*) and plants (*Vicentini et al., 2023*) support the external coincidence model (*Pittendrigh and Minis, 1964*). This model explains how the circadian clock regulates the expression of a light-sensitive substance during the late dark phase. Under long days, this substance is degraded by morning light, suppressing a winter response, while the substance accumulates under short days (and long nights), inducing a winter-like response. In the case of krill, this response could include sexual regression, increased lipid accumulation (*Kawaguchi et al., 2007*; *Höring et al., 2018*), and a reduction in overall metabolism (*Meyer et al., 2010*; *Piccolin et al., 2018b*).

Our data on krill circadian swimming activity of animals sampled from the field in different seasons show a remarkably stable peak in late-night activity (*Figure 5e–h*). Assuming that the behavioral activity under constant darkness reflects the direct influence of the circadian clock, our findings suggest that the krill circadian clock can measure the length of the day (or night) via external coincidence. The increased rhythmicity and synchronization of animals under shortening photoperiods indicate the potential of precise day length measurement especially during seasonal physiological changes (*Meyer et al., 2010*). The day length information could subsequently be used directly to synchronize physiological functions with the environment and/or indirectly to entrain a circannual clock (*Miyazaki, 2023*), which subsequently drives rhythmic annual output. While the individual activity recordings provide first insights into potential mechanisms for the control of daily and seasonal processes in krill, it must be noted that behavioral output does not always reflect the detailed characteristics of the circadian oscillator (*Häfker et al., 2024*). Further investigations into the molecular clock mechanism and its downstream functions across seasons are needed to clarify the involvement of the circadian clock in photoperiodism and circannual rhythms in krill.

Circadian clocks have the potential to anticipate the daily light-dark cycle and thereby enable a fine-tuned synchronization of behavioral and physiological functions with environmental cycles, which increases an organism's fitness (*Krittika and Yadav, 2020*). Consequently, synchronizing krill behavioral output with the day-night cycle provides potential adaptive advantages on various levels.

DVM of zooplankton seems to be a trade-off between predation risk and feeding (*Ringelberg and Van Gool, 2003*). As it is complex to directly sense predation risk, light is a robust proxy (*Benoit Bird and Moline, 2021*) and a circadian clock provides an additional mechanism to anticipate and synchronize the times of vertical migration, active feeding, and foraging with nighttime. Our data show that light, in addition to the clock, ensures robust, synchronized activity patterns. Differences between entrained and free running conditions further indicate that evening and late-night activity are under clock control. At the same time, the activity peak in the morning is a response to the onset of light. A differential regulation of certain phases of DVM has previously been suggested for marine zooplankton species (*Häfker et al., 2017*; *Cohen and Forward, 2005*). A clock-driven evening ascent would allow for the anticipation of sunset and thus the timely initiation of migration when krill swarms are in deeper water layers, resulting in optimized surface feeding time. The clock-controlled late-night activity could support a second feeding bout and preparation for a timely morning descent before sunrise increases visual predation risk. Interestingly, the decrease in clock-controlled swimming activity during the early night, right after the evening activity bout, may further facilitate a phenomenon called 'midnight sinking', which describes the sinking of animals to intermediate depths after the evening ascent, followed by a second rise to the surface before the morning descend. This behavior has been observed in a number of zooplankton species, including calanoid copepods (see *Tarling et al., 2002*; *Cushing, 1951* and references therein) and krill (*Tarling and Johnson, 2006*). While previous

studies suggested several exogenous factors, such as satiation or predator presence, as drivers of the midnight sink (*Tarling et al., 2002*; *Cushing, 1951*), our study suggests that this pattern may be partly under endogenous control. In situ hydroacoustic observations have demonstrated increased swimming speeds during nighttime for both *M. norvegica* and *E. superba* (*Zhou and Dorland, 2004*; *Klevjer and Kaartvedt, 2011*). This increase is likely related to active foraging and feeding behavior, as higher swimming speeds increase the chance of encountering food particles and thus increase food uptake. Nevertheless, swimming in krill is energetically costly and can make up more than 70% of the total energy expenditure (*Kils, 1981*). Reducing swimming activity to the baseline level during the day thus represents a significant opportunity for energy conservation.

At high latitudes, biological clocks are exposed to extreme changes in photoperiod throughout the year, including phases of midnight sun and polar night. Many organisms inhabiting polar regions show arrhythmicity, at least during parts of the year, and potential mechanisms include direct impacts on the central oscillator and uncoupling between output functions and the circadian clock (*Bloch et al., 2013*). Previous investigations of krill clock gene expression in the laboratory indicated a weaker oscillation of the krill clock and metabolic output functions under simulated extreme photic conditions (i.e. LL and LD 3:21), compared to more balanced day-night cycles (*Piccolin et al., 2018a*). Data on the transcriptomic and organismic level further indicate that krill can flexibly adapt to the photic conditions at different latitudes (*Höring et al., 2021*; *Höring et al., 2018*). This aligns with our findings, where krill, sampled across a wide range of photoperiods, remain behaviorally synchronized with the local photoperiod under entrained conditions, but show reduced endogenous rhythmicity under DD after entrainment to long-day conditions. Krill inhabits a broad range of latitudes, including regions with clear day-night cycles and regions with polar night and midnight sun during the seasonal extremes (*Siegel et al., 2016*). At the same time krill is transported rapidly between regions via ocean currents (*Fach et al., 2006*), potentially exposing the same individuals to various photic conditions during their life span. A weak circadian clock in close interaction with the ambient light regime could thus allow krill to adopt the best strategy for the environmental conditions prevailing in the current location. For example, at high latitudes, the adaptive advantage of predator evasion with DVM diminishes in summer when the upper water column is illuminated constantly. In this case, arrhythmic behavior would allow flexibly exploiting food patches at any time of day, when the need for food is high to fuel reproductive processes (*Quetin and Ross, 2001*). Further, a study on the marine annelid *Platynereis dumerilii* recently highlighted a differential regulation of clock-controlled processes, where behaviorally arrhythmic individuals show increased rhythmicity in physiological output (*Häfker et al., 2024*), indicating the possibility of flexibly adapting timed processes to environmental conditions.

Adapting to a fluctuating environment involves significant seasonal physiological changes in krill to survive the harsh, food-scarce winter (*Meyer et al., 2010*; *Meyer, 2012*). Crucial processes, such as accumulating enough lipids, must begin long before winter, requiring an anticipatory mechanism. Recent evidence suggests that photoperiod and circannual timing drive these seasonal changes in krill physiology, using photoperiod as a proxy for seasonal progression (*Meyer et al., 2010*; *Brown et al., 2011*; *Höring et al., 2018*; *Teschke et al., 2007*; *Piccolin et al., 2018b*). Rapid warming in the Southern Ocean is observed to cause shifts in phytoplankton bloom timing (*Thomalla et al., 2023*), which may lead to temporal mismatches between photoperiod synchronized krill physiology and their primary food source, with the potential to reduce reproductive success (*Helm et al., 2013*) and negatively impact the krill-dependent ecosystem. Understanding the mechanistic basis of species adaptation and their flexibility is therefore crucial in times of rapid environmental change (*Helm et al., 2013*).

We used a new krill activity monitor to identify the endogenous underpinnings of swimming activity of wild-caught Antarctic krill. The results of our experiments provide clear evidence that a circadian clock underlies krill behavior, driving a distinct bimodal pattern of clock-controlled swimming activity. The damping of the activity amplitude under constant darkness suggests that krill possess a 'weak', highly flexible circadian clock adapted to a life under extreme and variable conditions. These findings are further supported by experiments conducted under simulated short-day and long-day conditions, demonstrating a flexible adaptation of the krill swimming activity pattern to a wide range of photoperiods. The results provide novel insights into the combined impacts of the circadian clock and light on the daily regulation of swimming activity. Furthermore, hydroacoustic recordings demonstrate that most krill swarms sampled exhibited synchronized DVM in the field in the days directly before sampling for behavioral experiments, indicating that in this region, krill remain behaviorally

synchronized across a wide range of photoperiods. Seasonal recordings of field-entrained circadian swimming activity suggest that the krill circadian clock may be involved in measuring day length and, consequently, the timing of seasonal events. The findings provide novel insights into the mechanistic underpinnings of daily and seasonal timing in Antarctic krill, a marine pelagic key species, endemic to a high-latitude region. Mechanistic studies are a prerequisite for understanding how krill adapt to their specific environment and their flexibility in responding to environmental changes.

## Materials and methods

### Animal collection

Sampling of Antarctic krill (*E. superba*) for experimental purposes was conducted at various points throughout the seasonal cycle from the Bransfield Strait and South Orkney Island regions (for detailed locations, please refer to *Table 1* and *Figure 1—figure supplement 1*) by use of the continuous fishing system onboard the krill fishing vessel Antarctic Endurance. During fishing, the vessel trawled at a speed of 1.5–2 knots using a commercial trawl. The krill was pumped on board by creating a vacuum in a hose connected to the cod-end of the trawl. On board, the krill was separated from the water on a metal grate, from which the krill was sampled. Sampled krill were kept in surface seawater at densities of ~1 Ind./L, at 1°C under constant darkness for an acclimation period between 4 and 10 hr, to reduce the impact of sampling stress and to check for the condition of krill individuals before transfer to the experimental setup. Prior to the start of each experiment, the experimental columns were filled with seawater sampled from the surface. For experiments conducted during summer, late summer, and autumn, when high food concentrations were to be expected, seawater was filtered through a 0.25 µm filter. During winter, field measurements showed a very low surface chlorophyll *a* level (below 0.5 µg/L; data not shown) and any potential larger particles in the water were excluded by sedimentation. The columns were distributed to the setup and the temperature was adjusted to 0.8°C. Only krill that were unharmed and lively were selected for behavioral experiments in the setup. At the end of each experiment, the krill were removed from the setup, evaluated for overall condition and total length (measured from the front of the eye to the tip of the telson, excluding setae), and their sex was determined under a stereomicroscope.

### Krill swimming activity recording and experimental design

Krill swimming activity was recorded using the AMAZE setup, described in detail in *Hüppe et al., 2024*. In short, the recording principle is based on krill swimming in vertical acrylic experimental columns (height: 80 cm, diameter: 9 cm). Each column contains five infrared (IR) detector modules, equally spaced over the height of the column. Vertical movements of krill are detected by IR light beam breaks. A computer on each column controls data acquisition and stores beam break data. The experimental columns are placed in a compressor cooled incubator, which allows for a precise temperature control. Programmable LED light bars in the top of the incubator simulate underwater light spectra and daily light intensity cycles.

To investigate the behavioral activity of individual krill under light-dark conditions, 12 krill were exposed to 3 days of simulated short-day conditions (short-day treatment), and 10 krill to 5 days of simulated long-day conditions (long-day treatment). Due to technical problems, only data from 11 krill were analyzed from experiment 1 and data from 9 krill from experiment 2. Light-dark cycles were adjusted to a photoperiod of 5.5 hr simulating short days and 15 hr simulating long days. Day length was arranged symmetrically around local solar noon, to approximate the natural photoperiodic light regime at the time of sampling. The intensity and spectrum in the activity monitor were adjusted to field measurements, as described in *Hüppe et al., 2024*. In short, light regime increased and decreased in a linear fashion from a maximum light intensity of 8.8 mW/m$^2$ during midday. To further investigate the involvement of an endogenous clock in krill activity, the initial phase of day-night simulations was followed by 5 days of constant darkness conditions in both experiments.

To investigate seasonal changes in krill circadian activity, four additional experiments were conducted, namely summer treatment (sampling date: January 22, 2022), late summer treatment (sampling date: March 1, 2022), autumn treatment (sampling date: March 15, 2022), and winter treatment (sampling date: June 10, 2022). For these experiments, krill were caught at the dates described

above and after acclimation, 10 animals per experiment were distributed to the activity monitor, where their activity was recorded for 4–8 days under constant darkness conditions (see *Table 1* for details).

## Behavioral data analysis

### Experiment 1 (LD-DD)

Swimming activity of krill individuals was calculated from raw beam breaks as described in *Hüppe et al., 2024*. In short, we only considered upward swimming movements of individuals to separate baseline activity from increased activity, as krill are negatively buoyant. To accomplish this, we organized the raw beam break data from all five detector modules in each experimental column in chronological order. We selected only those beam break detections that occurred after a detection in the detector module positioned lower on the column. Like this, we consider upward swimming movements throughout the full height of the column. Beam breaks caused by upward swimming were summed over 10 min intervals and normalized between 0 and 1 for each individual. Data were smoothed by a centered moving average, with a smoothing window of 6 data points under light-dark simulations and 24 data points under constant darkness conditions.

To calculate the group activity of rhythmic krill under DD conditions, we first detrended the normalized swimming activity data of each individual to account for individual variability in baseline shifts of activity over time. This was done by subtracting the daily mean activity from each activity value of the same day. As the individual FRP varies between individuals, in the next step we corrected the data for each individual's FRP and assigned them to a 24 hr day. To achieve this, we tested the detrended and smoothed activity data of each krill for rhythmic activity and determined the period of the rhythm using the Lomb-Scargle periodogram analysis. We considered individuals with a period $\tau$ between 20 and 28 hr, a power of ≥50, and a p-value of less than 0.01 to be significantly rhythmic. The activity data of each rhythmic krill were subsequently assigned a new modified timestamp to cover the individual FRP within a 24 hr experimental day. This maintains the original resolution of the time series data while allowing for comparative analysis of individuals with differing FRPs. Group behavior was determined by calculating the mean and standard error of the mean (s.e.m.) per 10 min time interval and subsequent smoothing, as described above.

Average day analysis for a group of krill under LD cycles was done by determining the mean and s.e.m. per 10 min time interval of a 24 hr day over all experimental days and all individuals in one experiment and subsequent smoothing. Average day analysis for a group of krill under DD conditions was done in the same way, but based on the detrended and FRP-corrected data. The amplitude of swimming activity under LD and DD conditions was based on individual, normalized, detrended, and smoothed activity data, and was calculated for every experimental day as the difference between the daily maximal and minimal swimming activity. Differences in the activity amplitude between light conditions (i.e. LD vs. DD) were tested with the Mann-Whitney U-test and amplitude differences between the first and last day under DD conditions were tested with a paired t-test (R *stats* package version 4.1.2). All statistical analysis assumed a significance level of $p > 0.05$.

### Experiment 2 (DD)

Group activity and average day activity for seasonal experiments (experiment 2, summer, late summer, autumn, and winter treatment) was done as for DD conditions described above, but only the first 4 days of activity of each experiment were considered. This was to allow for a better comparison and to account for the effect of damping rhythms after several days under DD.

All data handling, analysis, and visualization was done with the *R programming language* (version 4.1.2, *R Development Core Team, 2021*) in *RStudio* (version 2023.12.1.402), using the *tidyverse* package (version 2.0.0, *Wickham et al., 2019*). Rhythm analysis and period estimation was done with the R package *lomb* (*Ruf, 1999*); local sun data (i.e. sunset and sunrise) were retrieved from the *suncalc* package (version 0.5.1).

## Hydroacoustic data recording and visualization

Hydroacoustic data were collected using a hull-mounted SIMRAD ES80 echosounder (Kongsberg Maritime AS) aboard the Antarctic Endurance, covering 3 days before the sampling for each of the seasonal behavioral experiments of experiment 2. The signal received from the 200 kHz band was used to visualize the vertical distribution of krill swarms beneath the ship. The raw acoustic data

from the summer, late summer, and autumn periods were converted to mean volume backscattering strength and binned to a time resolution of 1 s and depth bins of 0.5 m using Echopype (*Lee, 2021*). For the period of winter, raw acoustic data were not available. An alternative approach was employed, utilizing a novel method proposed by *Bahlburg et al., 2023a*, to reconstruct the backscattering signal from a dataset of screenshots displaying the visualized hydroacoustic signal of the echosounder. We only included data during active fishing periods and the vessel is specifically targeting *E. superba*, which occurs in large monospecific aggregations. Further, krill fishery bycatch rates are very low (0.1–0.3%, *Krafft et al., 2023*), which makes it highly probable that the recorded signal represents krill swarms. Data handling and visualization were done with R in RStudio, using the packages *tidyverse* and *scico* (version 1.3.1).

## Acknowledgements

We would like to thank the captain and crew of the Antarctic Endurance and Aker Biomarine for logistical and technical support during our field campaigns, Sara Driscoll for support during the experiments, and Nils Reinhard, Dirk Rieger, and Laura Payton for valuable discussions.

## Additional information

### Funding

| Funder | Grant reference number | Author |
| --- | --- | --- |
| Deutsche Forschungsgemeinschaft | FO 207/17-1 | Lukas Hüppe |

The funders had no role in study design, data collection and interpretation, or the decision to submit the work for publication.

### Author contributions

Lukas Hüppe, Conceptualization, Data curation, Formal analysis, Visualization, Methodology, Writing – original draft, Writing – review and editing; Dominik Bahlburg, Visualization, Methodology, Writing – review and editing; Ryan Driscoll, Methodology, Writing – review and editing; Charlotte Helfrich-Förster, Conceptualization, Supervision, Funding acquisition, Methodology, Project administration, Writing – review and editing; Bettina Meyer, Conceptualization, Supervision, Funding acquisition, Project administration, Writing – review and editing

### Author ORCIDs

Lukas Hüppe ⬛ https://orcid.org/0000-0002-7793-9046
Dominik Bahlburg ⬛ https://orcid.org/0000-0003-0210-0649
Bettina Meyer ⬛ https://orcid.org/0000-0001-6804-9896

Reviewer #1 (Public review): https://doi.org/10.7554/eLife.103096.3.sa1
Reviewer #2 (Public review): https://doi.org/10.7554/eLife.103096.3.sa2
Author response https://doi.org/10.7554/eLife.103096.3.sa3

## Additional files

### Supplementary files

MDAR checklist

### Data availability

The datasets and analysis scripts used for the analysis of krill swimming activity, as well as of hydroacoustic recordings are available in Zenodo.

The following dataset was generated:

| Author(s) | Year | Dataset title | Dataset URL | Database and Identifier |
|---|---|---|---|---|
| Hüppe L, Bahlburg D, Driscoll R, Helfrich-Förster C, Meyer B | 2025 | Data set and analysis code to reproduce figures presented in "A circadian clock drives behavioral activity in Antarctic krill (Euphausia superba) and provides a potential mechanism for seasonal timing" (Hüppe et al. 2025) | https://zenodo.org/records/14766951 | Zenodo, 10.5281/zenodo.14766951 |

The following previously published dataset was used:

| Author(s) | Year | Dataset title | Dataset URL | Database and Identifier |
|---|---|---|---|---|
| Hüppe L, Bahlburg D, Busack M, Lemburg J, Payton L, Reinhard N, Rieger D, Helfrich-Förster C, Meyer B | 2024 | Data set and analysis code to reproduce figures presented in "A new Activity Monitor for Aquatic Zooplankter (AMAZE) allows the recording of swimming activity in wild-caught Antarctic krill (Euphausia superba)" (Hüppe et al. 2024) | https://zenodo.org/records/12792874 | Zenodo, 10.5281/zenodo.12792874 |

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
