## [Editor Report · eLife Assessment]

This **important** study substantially advances our understanding of the circadian clock in Antarctic krill, a key species in the Southern Ocean ecosystem. Through logistically challenging shipboard experiments conducted across seasons, the authors provide **compelling** evidence for their conclusions. The study will be of broad interest to marine biologists and ecologists.

---

## [Referee Report · Reviewer #1 (Public review)]

Hüppe and colleagues had already developed an apparatus and an analytical approach to capture swimming activity rhythms in krill. In a previous manuscript they explained the system, and here they employ it to show a circadian clock, supplemented by exogenous light, produces an activity pattern consistent with "twilight" diel vertical migration (DVM; a peak at sunset, a midnight sink, and a peak in the latter half of the night).

They used light:dark (LD) followed by dark:dark (DD) photoperiods at two times of the year to confirm the circadian clock, coupled with DD experiments at four times a year to show rhythmicity occurs throughout the year along with DVM in the wild population. The individual activity data show variability in the rhythmic response, which is expected. However, their results showed rhythmicity was sustained in DD throughout the year, although the amplitude decayed quickly. The interpretation of a weak clock is reasonable, and they provide a convincing justification for the adaptive nature of such a clock in a species that has a wide distributional range and experiences various photic environments. These data also show that exogenous light increases the activity response and can explain the morning activity bouts, with the circadian clock explaining the evening and late-night bouts. This acknowledgement that vertical migration can be driven by multiple proximate mechanisms is important.

The work is rigorously done, and the interpretations are sound. I see no major weaknesses in the manuscript. Because a considerable amount of processing is required to extract and interpret the rhythmic signals (see Methods and previous AMAZE paper), it is informative to have the individual activity plots of krill as a gut check on the group data.

The manuscript will be useful to the field as it provides an elegant example of looking for biological rhythms in a marine planktonic organism and disentangling the exogenous response from the endogenous one. Furthermore, as high-latitude environments change, understanding how important organisms like krill have the potential to respond will become increasingly important. This work provides a solid behavioral dataset to complement the earlier molecular data suggestive of a circadian clock in this species.

---

## [Referee Report · Reviewer #2 (Public review)]

Summary:

This manuscript provides experimental evidence on circadian behavioural cycles in Antarctic krill. The krill were obtained directly from krill fishing vessels and the experiments were carried out on board using an advanced incubation device capable of recording activity levels over a number of days. A number of different experiments were carried out where krill were first exposed to simulated light:dark (L:D) regimes for some days followed by continuous darkness (DD). These were carried out on krill collected during late autumn and late summer. A further set of experiments was performed on krill across three different seasons (summer, autumn, winter), where incubations were all DD conditions. Activity was measured as the frequency by which an infrared beam close to the top of the incubation tube was broken over unit time. Results showed that patterns of increased and decreased activity that appeared synchronised to the LD cycle persisted during the DD period. This was interpreted as evidence of the operation of an internal (endogenous) clock. The amplitude of the behavioural cycles decreased with time in DD, which further suggests that this clock is relatively weak. The authors argued that the existence of a weak endogenous clock is an adaptation to life at high latitudes since allowing the clock to be modulated by external (exogenous) factors is an advantage when there is a high degree of seasonality. This hypothesis is further supported by seasonal DD experiments which showed that the periodicity of high and low activity levels differed between seasons.

Strengths:

Although there has been a lot of field observations of various circadian type behaviour in Antarctic krill, relatively few experimental studies have been published considering this behaviour in terms of circadian patterns of activity. Krill are not a model organism and obtaining them and incubating them in suitable conditions are both difficult undertakings. Furthermore, there is a need to consider what their natural circadian rhythms are without the overinfluence of laboratory-induced artefacts. For this reason alone, the setup of the present study is ideal to consider this aspect of krill biology. Furthermore, the equipment developed for measuring levels of activity is well-designed and likely to minimise artefacts.

---

## [Author Response]

The following is the authors’ response to the original reviews

**Reviewer #1 (Public review):**
Hüppe and colleagues had already developed an apparatus and an analytical approach to capture swimming activity rhythms in krill. In a previous manuscript they explained the system, and here they employ it to show a circadian clock, supplemented by exogenous light, produces an activity pattern consistent with "twilight" diel vertical migration (DVM; a peak at sunset, a midnight sink, and a peak in the latter half of the night).They used light:dark (LD) followed by dark:dark (DD) photoperiods at two times of the year to confirm the circadian clock, coupled with DD experiments at four times of year to show rhythmicity occurs throughout the year along with DVM in the wild population. The individual activity data show variability in the rhythmic response, which is expected. However, their results showed rhythmicity was sustained in DD throughout the year, although the amplitude decayed quickly. The interpretation of a weak clock is reasonable, and they provide a convincing justification for the adaptive nature of such a clock in a species that has a wide distributional range and experiences various photic environments. These data also show that exogenous light increases the activity response and can explain the morning activity bouts, with the circadian clock explaining the evening and late-night bouts. This acknowledgement that vertical migration can be driven by multiple proximate mechanisms is important.The work is rigorously done, and the interpretations are sound. I see no major weaknesses in the manuscript. Because a considerable amount of processing is required to extract and interpret the rhythmic signals (see Methods and previous AMAZE paper), it is informative to have the individual activity plots of krill as a gut check on the group data.The manuscript will be useful to the field as it provides an elegant example of looking for biological rhythms in a marine planktonic organism and disentangling the exogenous response from the endogenous one. Furthermore, as high latitude environments change, understanding how important organisms like krill have the potential to respond will become increasingly important. This work provides a solid behavioral dataset to complement the earlier molecular data suggestive of a circadian clock in this species.

We appreciate the positive evaluation of our work by Reviewer 1, acknowledging our approach to record locomotor activity in krill and the importance of the findings in assessing krill’s potential to respond to environmental change in their habitat.

**Reviewer #2 (Public review):**
Summary:This manuscript provides experimental evidence on circadian behavioural cycles in Antarctic krill. The krill were obtained directly from krill fishing vessels and the experiments were carried out on board using an advanced incubation device capable of recording activity levels over a number of days. A number of different experiments were carried out where krill were first exposed to simulated light:dark (L:D) regimes for some days followed by continuous darkness (DD). These were carried out on krill collected during late autumn and late summer. A further set of experiments was performed on krill across three different seasons (summer, autumn, winter), where incubations were all DD conditions. Activity was measured as the frequency by which an infrared beam close to the top of the incubation tube was broken over unit time. Results showed that patterns of increased and decreased activity that appeared synchronised to the LD cycle persisted during the DD period. This was interpreted as evidence of the operation of an internal (endogenous) clock. The amplitude of the behavioural cycles decreased with time in DD, which further suggests that this clock is relatively weak. The authors argued that the existence of a weak endogenous clock is an adaptation to life at high latitudes since allowing the clock to be modulated by external (exogenous) factors is an advantage when there is a high degree of seasonality. This hypothesis is further supported by seasonal DD experiments which showed that the periodicity of high and low activity levels differed between seasons.StrengthsAlthough there has been a lot of field observations of various circadian type behaviour in Antarctic krill, relatively few experimental studies have been published considering this behaviour in terms of circadian patterns of activity. Krill are not a model organism and obtaining them and incubating them in suitable conditions are both difficult undertakings. Furthermore, there is a need to consider what their natural circadian rhythms are without the overinfluence of laboratory-induced artefacts. For this reason alone, the setup of the present study is ideal to consider this aspect of krill biology. Furthermore, the equipment developed for measuring levels of activity is well-designed and likely to minimise artefacts.

We would like to thank Reviewer 2 for their positive assessment of our approach to study the influence of the circadian clock on krill behavior. We are delighted, that Reviewer 2 found our mechanistic approach in understanding daily behavioral patterns of Antarctic krill using the AMAZE set-up convincing, and that the challenging circumstances of working with a polar, non-model species are acknowledged.

WeaknessesI have little criticism of the rationale for carrying out this work, nor of the experimental design. Nevertheless, the manuscript would benefit from a clearer explanation of the experimental design, particularly aimed at readers not familiar with research into circadian rhythms. Furthermore, I have a more fundamental question about the relationship between levels of activity and DVM on which I will expand below. Finally, it was unclear how the observational results made here related to the molecular aspects considered in the Discussion.(1) Explanation of experimental design - I acknowledge that the format of this particular journal insists that the Results are the first section that follows the Introduction. This nevertheless presents a problem for the reader since many of the concepts and terms that would generally be in the Methods are yet to be explained to the reader. Hence, right from the start of the Results section, the reader is thrown into the detail of what happened during the LD-DD experiments without being fully aware of why this type of experiment was carried out in the first place. Even after reading the Methods, further explanation would have been helpful. Circadian cycle type research of this sort often entrains organisms to certain light cycles and then takes the light away to see if the cycle continues in complete darkness, but this critical piece of knowledge does not come until much later (e.g. lines 369-372) leaving the reader guessing until this point why the authors took the approach they did. I would suggest the following (1) that more effort is made in the Introduction to explain the exact LD/DD protocols adopted (2) that a schematic figure is placed early on in the manuscript where the protocol is explained including some logical flow charts of e.g. if behavioural cycle continues in DD then internal clock exists versus if cycle does not continue in DD, the exogenous cues dominate - followed by - major decrease in cyclic amplitude = weak clock versus minor decrease = strong clock and so on

We want to thank Reviewer 2 for pointing out that the experimental design and its rationale are not becoming clear early in the manuscript, especially for people outside the field of chronobiology. We added a new figure (now Fig. 1), illustrating the basic principle of chronobiological study design and how we adopted it. We also extended the description at the beginning of the Results section to clarify the rationale behind the experimental design.

(2) Activity vs kinesis - in this study, we are shown data that (i) krill have a circadian cycle - incubation experiments; (ii) that krill swarms display DVM in this region - echosounder data (although see my later point). My question here is regarding the relationship between what is being measured by the incubation experiments and the in situ swarm behaviour observations. The incubation experiments are essentially measuring the propensity of krill to swim upwards since it logs the number of times an individual (or group) break a beam towards the top of the incubation tube. I argue that krill may be still highly active in the rest of the tube but just do not swim close to the surface, so this approach may not be a good measure of "activity". Otherwise, I suggest a more correct term of what is being measured is the level of "upward kinesis". As the authors themselves note, krill are negatively buoyant and must always be active to remain pelagic. What changes over the day-night cycle is whether they decide to expend that activity on swimming upwards, downwards or remaining at the same depth. Explaining the pattern as upward kinesis then also explains by swarms move upwards during the night. Just being more active at night may not necessarily result in them swimming upwards.

We believe there is a slight misunderstanding in how what we call “activity” is measured. The experimental columns are equipped with five detector modules, evenly distributed over the height of the column. In our analysis we count all beam breaks caused by upward movement, i.e. every time a detector module is triggered after a detector module at a lower position has been triggered, and not only when the top detector module is triggered. In this way, we record upward swimming movements throughout the column, and not only when the krill swims all the way to the top of the column. This still means that what we are measuring is swimming activity, caused by upward swimming. We use this measure, to deliberately separate increased swimming activity, from baseline activity (i.e. swimming, which solely compensates for negative buoyancy) and inactivity (i.e. passive sinking).

Higher activity is thus at first interpreted as an increase in swimming activity, which in the field may result in upwards-directed swimming but also could mean a horizontal increase in activity, for example, representing increased foraging and feeding activity. This would explain the daily activity pattern observed under LD cycles (now Fig. 3), which shows a general increase in activity during the dark phase. This nighttime increase could be used for both upward directed migration during sunset and horizontal directed swimming for feeding and foraging throughout the night.

We added the following sentence to the description of the activity metric in the Methods section to clarify this point (lines 465-469):

“To accomplish this, we organized the raw beam break data from all five detector modules in each experimental column in chronological order. We selected only those beam break detections that occurred after a detection in the detector module positioned lower on the column. Like this, we consider upward swimming movements throughout the full height of the column.”

(3) Molecular relevance - Although I am interested in molecular clock aspects behind these circadian rhythms, it was not made clear how the results of the present study allow any further insight into this. In lines 282 to 284, the findings of the study by Biscontin et al (2017) are discussed with regard to how TIM protein is degraded by light via the clock photreceptor CRYTOCHROME 1. This element of the Discussion would be a lot more relevant if the results of the present study were considered in terms of whether they supported or refuted this or any other molecular clock model. As it stands, this paragraph is purely background knowledge and a candidate for deletion in the interest of shortening the Discussion.

We agree that this part is not directly related to the data presented in the manuscript. We, therefore, omitted this part in the revised version of the manuscript to keep the discussion concise and focused on the results.

Other aspects(i) 'Bimodal swimming' was used in the Abstract and later in the text without the term being fully explained. I could interpret it to mean a number of things so some explanation is required before the term is introduced.

We thank the Reviewer for pointing this out. We provided an explanation for the term “bimodal” in the Results section, where the two clock driven activity bouts are described first, by extending the sentence in lines 161-164, which now reads:

“This suggests that the circadian clock drives a distinct bimodal activity pattern with two activity peaks in one day, i.e. the evening and late-night activity bouts, while. In contrast, the morning activity bout is triggered by the onset of illumination in the experimental set-up.”.

(ii) Midnight sinking - I was struck by Figure 2b with regards to the dip in activity after the initial ascent, as well as the rise in activity predawn. Cushing (1951) Biol Rev 26: 158-192 describes the different phases of a DVM common to a number of marine organisms observed in situ where there is a period of midnight sinking following the initial dusk ascent and a dawn rise prior to dawn descent. Tarling et al (2002) observe midnight sinking pattern in Calanus finmarchicus and consider whether it is a response to feeding satiation or predation avoidance (i.e. exogenous factors). Evidence from the present study indicates that midnight sinking (and potential dawn rise) behaviour could alternatively be under endogenous control to a greater or lesser degree. This is something that should certainly be mentioned in the Discussion, possibly in place of the molecular discussion element mentioned above - possibly adding to the paragraph Lines 303-319.

We would like to thank the Reviewer for pointing this out and agree that adding the idea of an endogenous control of midnight sinking would be interesting to the discussion. We added the following section to the Discussion (lines 335-343):

“Interestingly, the decrease in clock-controlled swimming activity during the early night, right after the evening activity bout, may further facilitate a phenomenon called “midnight sinking”, which describes the sinking of animals to intermediate depths after the evening ascent, followed by a second rise to the surface before the morning descend. This behavior has been observed in a number of zooplankton species, including calanoid copepods (see 69, 70 and references therein) and krill (71). While previous studies suggested several exogenous factors, such as satiation or predator presence, as drivers of the midnight sink (69, 70), our study suggests that this pattern may be partly under endogenous control.”

(iii) Lines 200-207 - I struggled to follow this argument regarding Piccolin et al identifying a 12 h rhythm whereas the present study indicates a ~24 h rhythm. Is one contradicting the other - please make this clear.

In our study, we found that the circadian clock drives a bimodal pattern of swimming activity in krill, meaning it controls two bouts of activity in a 24-hour cycle. Piccolin et al. (2020) identified a swimming activity pattern of ~12 h (i.e. two peaks in 24 h) at the group level, which aligns with our findings at the individual level. We revised the Section in the discussion for more clarity, which now reads:

“Data from Piccolin et al. (20) showed a strong damping of the amplitude and indication of a remarkably short (~12 h) free running period (FRP) of vertical swimming behavior of a group of krill under constant darkness (20). The short period found in Piccolin et al. (20) complements is in line with our findings of a bimodal activity pattern the pattern of swimming activity under DD conditions on the individual level found in the present study, suggesting that the ~12 h rhythm in group swimming behavior in Piccolin et al. (20) could have resulted from a bimodal activity pattern at the individual level, as found in our study.” (lines 212-219).

(iv) Although I agree that the hydroacoustic data should be included and is generally supportive of the results, I think that two further aspects should be made clear for context (a) whether there was any groundtruthing that the acoustic marks were indeed krill and not potentially some other group know to perform DVM such as myctophids (b) how representative were these patterns - I have a sense that they were heavily selected to show only ones with prominent DVM as opposed to other parts of the dataset where such a pattern was less clear - I am aware of a lot of krill research where DVM is not such a clear pattern and it is disingenuous to provide these patterns as the definitive way in which krill behaves. I ask this be made clear to the reader (note also that there is a suggestion of midnight sinking in Fig 5b on 28/2).

To clarify the mentioned points concerning the hydroacoustic data:

a) As mentioned in the Methods section, only hydroacoustic data during active fishing was included in the analysis. *E. superba* occurs in large monospecific aggregations, and the fishery actively targets *E. superba* and monitors their catch and the proportion of non-target species continuously with cameras. Krill fishery bycatch rates are very low (0.1–0.3%, Krafft et al. 2022), and fishing operations would stop if non-target species were caught in significant proportions at any time. Therefore, and supported by our own observations when we conducted the experiments, we argue that it is a valid assumption that *E. superba* predominantly causes the backscattering signal shown in Figure 5 (now Fig. 6).

b) We are aware of the fact that DVM patterns of Antarctic krill are highly variable and that normal DVM patterns do not need to be the rule (e.g. see our cited study on the plasticity of krill DVM by Bahlburg et al. 2023). The visualized data were not selected for their DVM pattern but represent the period directly preceding the sampling for behavioral experiments in four seasons (experiment 2), including the day of sampling. These periods were chosen to assess the DVM behavior of krill swarms in the field in the days before and during the sampling for behavioral experiments.

To improve understanding, we modified the description in the Results, Discussion, and Methods sections, as well as the caption of Figure 5 (now Fig. 6), which now read:

“To investigate whether krill swarms exhibited daily behavioral patterns in swimming behavior in the field before they were sampled for seasonal experiments, hydroacoustic data were recorded from the fishing vessel, continuously over a three-day period prior to sampling for the seasonal experiments described above…” (lines 191-194).

“Furthermore, hydroacoustic recordings demonstrate that most krill swarms sampled exhibited synchronized DVM in the field in the days directly before sampling for behavioral experiments, indicating that in this region, krill remain behaviorally synchronized across a wide range of photoperiods.” (lines 397-400).

“Hydroacoustic data were collected using a hull-mounted SIMRAD ES80 echosounder (Kongsberg Maritime AS) aboard the Antarctic Endurance, covering three days before the sampling for each of the seasonal behavioral experiments of experiment 2” (lines 512-515).

“We only included data during active fishing periods and the vessel is specifically targeting *E. superba*, which occurs in large monospecific aggregations. Further, krill fishery bycatch rates are very low (0.1-0.3%, 84), which makes it highly probable that the recorded signal represents krill swarms.” (lines 523-526).

“Hydroacoustic recordings showing the vertical distribution of krill swarms in the upper water column (<220 m) below the vessel, visualized by the mean volume backscattering signal (200 kHz), on the three days prior to krill sampling for experiments…” (lines 802-804).

**Recommendations for the authors:**

**Reviewer #1 (Recommendations for the authors):**
As noted in the public review, this is a logical and well-written manuscript. I have very few comments to consider addressing.The Results lead with a paragraph outlining the experimental approach. This is good, but you use the term "experiments" to refer to both the two sets, and the two or four subsets of experiments. Perhaps consider the subset experiments as "treatments"? I understood what you meant, but it took a few read-throughs to be sure I got it.

We thank the reviewer for pointing this out and changed the nomenclature of the experiments throughout the manuscript. We now refer to the two sets of experiments as experiment 1 and 2, to the subsets of experiment 1 as “short day treatment” and “long day treatment”, and to the subsets of experiment 2 as summer treatment, late summer treatment, autumn treatment, and winter treatment. We also believe that the new Figure 1 is now helping to follow the experimental design more efficiently.

Ln 140: "...off and decrease at lights-on."

We adjusted the sentence accordingly.

Ln 244: Can you define "extreme photic conditions"? I get what you mean, but to be clear to the reader this would help.

We adjusted the sentence, which now reads:

“This could confer a significant adaptive advantage to species inhabiting environments characterized by extreme photic conditions (53, 54, 60), such as phases of polar night or midnight sun as well as rapid changes in daylength, or species that rely on precise photoperiodic time measurement for accurate seasonal adaptation.” (lines 258-261).

Figures: Consider adding an LSP for groups in Fig 1. Also, it would be useful to have LSP period estimates for each individual tested. This could be a separate table, or it could be added to the individual activity plots. Should S3 and S4 be reversed?

We thank the reviewer for their suggestion and added an LSP as figure 1d (now Fig. 2d) to statistically support the group activity shown in Figure 1c (now Fig. 2c) as suggested. We added the individual animals' LSP period estimates to supplementary figures S2, S7, S8, S9, and S10. We also reversed Figures S3 and S4 to match the appearance in the main text.

Fig 5: are the light regime bars for b and c correct? They look similar, but there are only 15 days apart, so perhaps they are correct as is.

We double checked the light regime bars in Fig. 5b and c (now 6b and c) and they are correct as is.